# Utilization of technology to provide on-the-job trainings on Emergency Obstetric and Neonatal Care: Perspectives of nurses and midwives working in Rwanda's remote health facilities

Thierry Claudien Uhawenimana[1]*, Mathias Gakwerere[2], Anaclet Ngabonzima[3], Assumpta Yamuragiye[4], Florien Harindimana[5], Jean Pierre Ndayisenga[6]

1 School of Nursing and Midwifery, College of Medicine and Health Sciences, University of Rwanda, Kigali, Rwanda, 2 United Nations Populations Fund East and Southern Africa Region, Johannesburg, South Africa, 3 JSI Research and Training Institute Inc, Denver, Colorado, United States of America, 4 School of Health Sciences, College of Medicine and Health Sciences, University of Rwanda, Kigali, Rwanda, 5 United Nations Population Fund, Kigali, Rwanda, 6 Arthur Labatt Family School of Nursing, Faculty of Health Sciences, Western University, London, Ontario, Canada

* tcuhawenimana@gmail.com

## Abstract

### Introduction

One of the targets for the third sustainable development goals is to reduce worldwide maternal mortality ratio (MMR) to less than 70 deaths per 100,000 live births by 2030. To address issues affecting women and the newborns during childbirth and postnatal period, concerted efforts from governments and their stakeholders are crucial to maximize the use of technology to enhance frontline health professionals' skills to provide the emergency obstetric and newborn care (EmONC). However, no study has garnered nurses' and midwives' perspectives regarding the application of technology-enhanced learning approach to provide on-the-job Continuous Professional Development (CPD) and factors that may influence the application of this training approach in the Rwandan context.

### Methods

The study collected data from nurses and midwives from forty (40) public health facilities in remote areas nationwide. The study applied a qualitative descriptive design to explore and describe nurses' and midwives' perspectives on the feasibility and acceptability of technology enhanced learning approaches such as e-learning, phone-based remote training, and other online methods to provide trainings in EmONC. Two focus group discussions with EmONC mentors, two with nurses and midwives were conducted. Twelve key informant interviews were conducted. Participants were selected purposively. In total, 54 individuals were included in this study. A thematic approach was used to analyse data.

**Data Availability Statement:** The data underlying the results presented in the study are available without any restrictions. We have provided

Kinyarwanda interview transcripts together with this manuscript.

**Funding:** the authors received no specific funding for this work.

**Competing interests:** Authors declare that there are no competing interests. The views expressed in this paper belong to the authors and do not necessarily align the views of UNFPA or the United Nations and the authors' affiliated institutions.

## Results

Nurses and midwives highlighted the need to provide refresher trainings about the management of pre-eclampsia. Most of the EmONC trainings are still provided face-to-face and the use of technology enhanced learning approaches have not yet been embraced in delivering EmONC CPDs for nurses and midwives in remote areas. Nurses and midwives found the first developed prototype of smartphone app training of the EmONC acceptable as it met the midwives' expectations in terms of the knowledge and skills' gap in EmONC.

## Conclusion

Although the newly developed application was found acceptable, further research involving practical sessions by nurses and midwives using the developed application is needed to garner views about the ease of use of the application, relevance of the EmONC uploaded content on the app, and needed improvements on the app to address their needs in EmONC.

## Introduction

Globally, women and newborns continue to be at high risk of deaths and morbidities resulting from sub-optimal care during labour, childbirth, and postpartum period [1, 2]. The World Health Organization (WHO) reports that every day, around 817 women lose their lives as a result of preventable causes related to pregnancy and childbirth [2]. Although in the last two decades significant efforts have been made to reduce maternal mortality ratio, maternal deaths remain high in low- and middle-income countries (LMICs); particularly in the Sub-Saharan Africa (SSA) [3]. SSA alone accounts for roughly two-thirds (196,000) of all global maternal deaths [3]. The major causes of these deaths are obstetric complications such severe bleeding and infections occurring after childbirth and abortion, eclampsia and pre-eclampsia, and complications from delivery [4, 5].

Obstetric complications impacting a pregnant woman during and after childbirth can also lead to significant consequences for the neonate. For newborns, the first month of life is the most vulnerable period of their survival [6]. The WHO reports that 2.4 million newborns die every year and most of these deaths (75%) occur during the first week of life because of prematurity, birth asphyxia, infections, and other birth-related complications [6].

One of the targets for the third sustainable development goals (SDGs) is to reduce maternal mortality ratio (MMR) to less than 70 deaths per 100,000 live births by 2030 [7, 8]. To achieve this target, it is imperative to invest in interventions that optimize the women's care during pregnancy, childbirth, and postnatal period. Since most of the maternal and neonatal deaths and morbidities primarily result from causes that can be prevented if women get access to appropriate medical assistance during pregnancy and childbirth, the implementation of EmONC services can be one of the effective interventions to address the contributing factors to maternal and neonatal mortalities. Therefore, concerted efforts from governments and their stakeholders are crucial to enhance frontline health professionals' skills to provide EmONC [9]. In-service EmONC capacity building trainings need to build on evidence-based learning methods including technology enabled remote capacity building approaches [10, 11].

Rwanda is progressively making impressive strides in reducing maternal and neonatal deaths and morbidities. Currently, maternal mortality stands at 210/100,000 whereas neonatal mortality rate stands at 18/1,000 live births [12, 13]. Despite the progress made, Rwanda needs

to invest in interventions projects targeting at reducing the current maternal mortality rate to 70/100,000 live births set to achieve the sustainable development goals.

Although remote training methods have not been commonly practiced to training health-care providers in Rwanda in the past years, the country is encouraging e-learning technologies in several sectors, particularly in the health sector. In particular, remote training being cost-effective compared to traditional learning methods can provide space for easy scale-up and improvement of content availability [14]. In this regard, The United Nations Population Fund (UNFPA) Rwanda, in collaboration with the Rwanda's Ministry of Health (MoH), has devel-oped a novel approach to enhance the training of healthcare providers, with a specific empha-sis on the provision of EmONC services. Drawing on successful experiences from other countries, this innovative solution utilizes various platforms such as Interactive-Voice-Response (IVR) system, Unstructured Supplementary Service Data (USSD) system, and a smartphone application to deliver effective training remotely via phone. Despite prospects and potential of this pilot remote training initiative, there has been no research conducted to explore how nurses and midwives viewed the new alternative for CPD training approach. Fur-thermore, available research from Rwanda focused on health facility managers' perceptions on the use of e-learning in providing CPD for health providers' capacity building purposes, but the study did not explore nurses' and midwives' views on remote trainings to provide CPDs [15].

## Objective

This study sought to garner nurses' and midwives' perspectives on how the newly developed remote training technological tools can be utilized to deliver learner-centered EmONC CPD trainings for nurses and midwives, and to identify the factors that may affect the implementa-tion of this training approach in the Rwandan context.

## Methodology

### Study area

Data was collected from nurses and midwives from forty (40) public health facilities in rural and semi-urban areas of five provinces in Rwanda (Southern, Northern, Eastern, and Western provinces plus the City of Kigali) that provided EmONC services (see S1 Checkilst). Participat-ing facilities included health posts, health centres, and district hospitals located in remote areas of Rwanda. Findings presented in this paper consists of the qualitative component of a coun-trywide mixed methods study that sought to determine how nurses' and midwives' access CPD for their capacity building in EmONC provision.

### Study design

This study was descriptive qualitative design. Given that there is limited research about the use of technology enhanced learning to deliver EmONC capacity building in the Rwandan context, qualitative descriptive design enabled flexibility in data collection and analysis to garner rich comprehensive insights from nurses and midwives. This study was conducted between Octo-ber 2020 and October 2021.

### Study population

The target population for this study was nurses and midwives working in maternity services and who were providing EmONC at the time of data collection. The sample to include in the current study were drawn from 149 nurses and midwives who participated in the quantitative

components of the project titled 'User research for remote capacity building of health care providers on EmONC in Rwanda'. The study excluded all participants with work experience less than 6 months at a given health facility as they may not yet have become familiar with the facility systems and various training opportunities for EmONC.

## Sample size and sampling strategy

Rwanda consists of a total of five provinces, encompassing 30 administrative districts. In order to determine the participants for our study, we ensured that each province was represented in the study. We initially chose specific locations where the research would take place. Within each administrative province, one district hospital was chosen at random, resulting in a total of five district hospitals that also represented the five administrative districts. For the remaining districts where a district hospital was not selected, we randomly picked eleven health centers and ten health posts. Subsequently, participants were recruited from the designated health facilities.

To achieve data saturation, two separate focus group discussions (FGDs) were carried out. The first FGD involved EmONC mentors who were both nurses and midwives, while the second FGD included nurses and midwives who had not yet received training in EmONC. Each FGD consisted of ten participants. To ensure comprehensive data collection, the FGDs were complemented with key informant interviews (KIIs). In total, five KIIs were conducted with nurses, five with midwives, and two joint interviews were held with one nurse and one midwife working in maternity. The selection of participants for both the FGDs and KIIs was purposeful. The study included a total of 54 individuals, which was deemed sufficient to obtain valuable insights from nurses and midwives regarding the effectiveness of remote training approaches for EmONC CPDs provision.

## Data collection process

A topic guide with questions covering familiarity in using technological tools to access EmONC trainings, factors influencing technology enhanced learning among nurses and midwives, and views about the newly developed smartphone application for EmONC training was used to moderate focus group discussions. To garner views about the newly developed EmONC training materials featured into the app, we presented the components of the application to the team participating in key informant interviews then we asked what they thought about them. This was done in order to find out if the newly designed remote training tool could be acceptable and feasible for use among nurses and midwives working in remove areas. The same tool that was used during the collection of focus group discussion was also used to facilitate the key informant interviews. Due to Covid-19 pandemic containment measures that imposed restricted movements, the research team collected data virtually as a mitigation measure for the risks that would be involved in collecting data face-to-face. Virtual data collection using phone calls and google meet with the support of the BEmONC mentors who were based at the data collection site was performed. The research team collected data virtually. EmONC district-based mentors facilitated the research team to reach selected respondents and further assisted in recording notes that complemented those collected by the research team. We ensured that HCPs felt comfortable expressing their views on the issues around the current online methods of capacity building. An experienced moderator facilitated the discussion with the assistance of a note-taker using a pre-designed interview guide. All interviews were audio-recorded. In case a respondent refused to be audio-recorded, the discussions/ interviews were written down by the data collection assistants at the study sites.

## Data analysis

The audio recorded interviews were transcribed, cleaned and translated from Kinyarwanda to English, the local language in which the discussions were conducted. To ensure the accuracy of the text, moderators and field note-takers performed the transcription. The qualitative research experts performed quality checks of the transcribed and translated files, so that it accurately captures the information shared by respondents in its context. They were reading each transcript while listening to its full audio and correcting errors or reviewing technical terms.

A thematic analysis that followed Richie and Spencer framework analysis steps was applied [16, 17] to obtain themes related to how the newly developed remote training technological tools can be utilized to deliver learner-centered EmONC CPD trainings for nurses and midwives, and to identify the factors that may affect the implementation of this training approach in the Rwandan context. All five steps of qualitative data analysis including reading, interpreting, coding, reducing, and displaying were performed to ensure consistency within the data. Upon completion of transcription and synthesis of FGDs and interviews, we developed a codebook, which framed the key themes from the data. The codebook was used to systematically code a subset of FGDs and interviews transcripts. The research team (experienced moderators and note takers) and qualitative research experts discussed the fieldwork experience and preliminary results of fieldwork. From these discussions, a further list of codes was developed to have a final codebook that was used by qualitative research experts to analyze all qualitative data systematically. We imported the translated interviews into qualitative analysis Atlas Ti software for coding. After development of the code list, the qualitative research experts reviewed all transcripts and extracted quotes appropriate to each of the primary codes. Subsequently, where there were mismatch in the coding process and themes' generation, the research team reviewed them and discussed them to reach consensus. Finally, the research team sat to categorise themes according to the objectives they responded to; namely, i) technology use in providing learner-centered EmONC CPD trainings for nurses and midwives in remote health facilities, ii) factors that may affect the use of technology enhanced learning approach in the Rwandan context, and iii) acceptability of using e-learning technology to provide EmONC CPD trainings.

**Measures undertaken to ensure trustworthiness.**   To ensure the rigour in the conduct of this study, interviews were conducted by experienced qualitative researcher who, in addition to their experience received a training before data collection started. Study participants were selected from different health facilities to ensure credibility through data triangulation. Furthermore, member checking was conducted to share the initial findings with participants and ensure that the researchers' interpretation reflected the perspectives from participants and had an opportunity to clarify meaning and gather more data.

## Ethical consideration

This research was conducted after getting the ethical approval from Rwanda National Ethics Committee and the permission from Rwandan Ministry of Health (MoH) to conduct this study (Ref: NHRC/2021/PROT/036) in identified settings.

The research team (who carried out the study virtually) and data collectors (who were at the facilities) adhered to principles of confidentiality and ethics in data collection. No person's name (except for identification of data collector) was recorded on any of the interviews. Permission to enter each facility for data collection assistants to facilitate the notetaking for FGDs and interviews was requested from the director or staff in-charge of the health facility at the beginning of each visit. The data collection assistants carried with them official letters of

cooperation from the MoH, and district level offices. The data collection assistants who were on the field facilitated to communicate the study to participants by reading the participant information sheet and coordinated the seeking of consent from participants at the study sites prior to proceeding with the virtual discussion or interviews by the research team. No incentive was provided to participants hence the participation was totally voluntary. Respondents were able to withdraw their participation anytime during the interview. Since the data collection was done virtually, participants first confirmed their voluntary participation by first providing a verbal consent. They were also requested to sign and return hard copies of consent forms through the data collection assistants.

## Findings

The nurses and midwives who took part in the qualitative study were chosen from the respondents of the survey section of this project. We recognize the drawback of not documenting the sociodemographic data of the participants involved in the qualitative study. Nevertheless, comprehensive information regarding the demographic details of the individuals who participated in the survey can be found in S1 File.

Three major themes emerged from the findings. They encompassed i) perceptions on efficacy of in-person versus online training, ii) moderators of technology enhanced learning for EmONC CPD delivery, and iii) acceptability and feasibility of the newly developed technology enhanced learning prototype for EmONC training.

### Theme 1: Perceptions on efficacy of in-person versus online training

The respondents reflected on some of the EmONC CPD trainings they have attended but which were predominantly provided face-to-face. The majority of respondents attended trainings on helping mother survive (HMS) and helping baby breathe (HBB). The common finding among all interviewees is that they appreciated the organizers of these CPD trainings, as they included the practical sessions as reported by one of the respondents.

*"There was a training on EmONC we had two weeks ago. We were trained on helping baby breathe and help mother survive skills. We learned how to conduct newborn resuscitation especially the positioning of newborn as someone may position the baby in a wrong way and this worsens his/her conditions. You get that it would have been more challenging to demonstrate this through online training."*

(Midwife, Mukungu Health Centre, KII)

Her views were echoed by most of the nurses and midwives during the FGDs as reflected by one quote from a participating nurse.

*"I attended training on helping mother survive. This training helped me to learn how to manage postpartum haemorrhage, severe preeclampsia, and eclampsia. This helps me a lot in my daily work. Now I am confident in what I do. I know all the steps I can take to manage those cases thanks to the practical sessions we held."*

(Nurse, Mukungu Health Centre, FGD)

The majority of nurses and midwives preferred face-to-face trainings over online courses. They argued that physical interactions between CPD providers and learners enhances the mastery of some the topics requiring practical skills. The respondents mentioned that despite the

usefulness of online CPD trainings particularly during emergencies such as Covid-19 pandemic, in-person trainings were perceived effective in equipping nurses and midwives with knowledge and translational practical skills in EmONC as illustrated in this quote.

*"Even if we would have online CPD trainings, the face-to-face trainings are still very important. This is because face-to-face training helps learners to master the subject easily. Face-to-face learning should be maintained for some modules especially for new topics. It would be better for us to use face-to-face training because it helps people to understand courses better than with the online system."*

(Nurse,_Gatsata Health Centre, KII)

## Theme 2: Moderators of technology enhanced learning for EmONC CPD delivery

Most nurses and midwives reported that they found it challenging to access CPD credits online because of insufficient skills in IT and lack of continued access to the strong internet network. According to most participants, this issue affected their accumulation of required number of CPD credits in order to get their professional licenses renewed. The respondents suggested that CPD credits awarded through online system should be increased as it takes time to finish the modules then afterwards few credits are awarded as explained by one of the participants.

Furthermore, nurses and midwives reported some moderators of technology enhanced learning approaches for accumulation of EmONC CPD credits. Whilst face-to-face training was the most preferred, some participants were aware of the benefits they might gain when in using technology enhanced learning methods. Some of the benefits mentioned include acquiring updated skills and information on EmONC in a timely way, having access to different didactic materials including videos and slides, accessing help on some case management remotely. They also mentioned that digital based trainings are cheap and convenient for the continuity of work.

"*What motivates me is that you get access to many things in a short time. The second thing is that you can access lessons in any format you want like Videos, audio and slides. When you cannot read slides, you use videos that demonstrate a certain procedure you want to learn. The other thing is that you can easily exchange with others about cases without having to move for face-to-face meetings*".

(Midwife, Gatsata Health Centre, FGD)

Another nurse also added that: "*It is easy to learn online as it is quick, and I do it while I am at my working place. There are no transport fees required.*" (Midwife and EmONC Mentor, _Kibilizi District Hospital, KII)_

The majority of respondents underscored that online learning had a potential to equip them with needed knowledge and skills to enable them to manage correctly all EmONC cases they receive thereby improving the quality of care given to mothers, babies, and families. However, participants expressed that the number of topics they covered regarding EmONC was not sufficient. They suggested that in the EmONC training package, there is a need to consider topics such as ultrasound use, management of hypertensive disorders in pregnancy, management of eclampsia, and performing manual vacuum aspiration in management of abortion. Most respondents from health centers appealed for training in cervical tear management,

neonatal resuscitation in case of fetal distress, and management of Rhesus incompatibility in pregnancy it can go unnoticed quickly leading to repetitive abortions.

*"The topic I would also like to be trained on is ultrasound. I would also like those topics about the management of preeclampsia and eclampsia be kept in the training. I am not saying that we did not receive any training on them, but when we are dealing with real cases, sometimes, it is difficult for us".*

(Nurse and EmONC Mentor,_Busanza Health Center, KII)

Another determining factor for nurses' and midwives' use of technology for self-directed learning about EmONC was the motivation to constantly enhancing their knowledge and skills in so that they reach a level where they felt confident and comfortable to provide quality care to mothers and newborns successfully with minimal supervision. Some of the views of the participants are shown below:

*"For me, I want to attain a level whereby I would be able to manage any conditions that a mother and her babies might have to prevent death. This is saddening to lose a mother or a baby."*

(Midwife,_Mukungu Health Centre, KII)

The majority of the respondents mentioned that technology enhanced learning for nurses and midwives would be successful if those providing trainings target the appropriate time for their self-learning particularly when they are off duties such as during weekends and evening hours. A few of them said that they might secure some free time to use at work when they do not have patients to attend to, especially when they have a challenging case to manage. Another alternative time for learning suggested by the respondents is when they are on annual leave.

Most nurses and midwives mentioned that they have some level of experience to locate online information and resources related to EmONC. They revealed that at some point during their work, they visited websites such as Healthline and Medline to access needed information about EmONC. Participants also shared that they accessed information from their peers using the WhatsApp social media platform.

*"In terms of access to information, especially those related to EmONC, there are many groups we have created for this purpose. For example, there is a group for all people working in maternity where we share information, and this helps us. It is also used to share information about upcoming trainings on EmONC. We also discuss difficult cases on that group."*

(Midwife,_Kinyinya Health Centre,KII)

However, some nurses and midwives; particularly from upcountry, found it challenging to search and retrieve online information about EmONC, because for some of them, the exercise required some level of knowledge in digital literacy skills and mastering all EmONC concepts in order to search relevant information meeting their needs.

### Theme 3: Acceptability of the newly developed prototype e-learning app for EmONC training

Although the prototype self-learning phone application to deliver EmONC was a new concept for most of them, nurses and midwives expressed an interest in using it. They labelled it a

perfect platform that will facilitate them to obtain essential skills in addressing some of the challenges they face in their daily professional activities. Most of them found it as an updated platform on current EmONC protocol. One key component most participants appreciated regarding the developed application was the ease of use and the fact that courses uploaded on it were leading to CPD credits.

*"I think it is good, especially in terms of career development. As you said, it would be great if we could accumulate CPD credits after we have used it. This would be a good way of keeping us up to date."*

(Nurse, Busanza Health Centre, KII)

*"Personally, as a midwife, this application is needed. This is a good platform to help HCPs to stay up to date with the latest science. It will also help the HCPs to be able to manage obstetric emergencies."*

(Midwife, Rusizi Health Centre, KII)

All participating nurses and midwives expressed that the digital based self-learning application came in due time because self-directed learning approach fits their nature of work. They reported that a digital based learning method would enable them to get acquainted with emerging technological advances in healthcare provision and further expose them to updated evidence-based knowledge and skills to provide maternal and neonatal care.

*"The application is coming on time, and it is a good one. What I can add on top of what have been said, is that this application will help us to continue to provide better service to our clients, as there are times when we are overworked because some of us have gone to training. We will use it while we continue to work which will help us to provide better services leading to prevention of maternal and child deaths."*

(Midwife and EmONC mentor, Kibilizi District Hospital, FGD)

During the discussions, nurses and midwives proposed to also avail an offline option of this prototype in order to mitigate the connectivity issue that may affect its functionality. Another barrier participants reported was financial constraint to access internet bundles and smartphones for some nurses and midwives, which can reduce their interest in using the application to learn.

*"Nurses and midwives need the offline mode because often times when it comes to internet; some people find it expensive. Others find it unnecessary to buy internet bundles for home use as they reside in areas where they have poor connectivity. For these reasons, an offline mode is better."*

(Nurse, Gatsata Health Centre, KII)

*"Before mentioning the issue of buying internet, many nurses do not even have a smartphone. Even those who have smartphones, they are not able to buy internet bundles. Therefore, an offline mode would be ideal so that they use it when they are at home and then connect themselves when they are at work using office internet."*

(Midwife, Mukungu Health Centre, FGD)

In terms of ways to utilize this prototype, all participants suggested there should be a way of coordinating its use at each health facility. They expressed also the need of having regular mentorship visits so that they can ask questions on what the application was not able to clarify.

*"Using technology and we share a link for discussing what we do not understand well in the prototype as we are doing today would be great. If possible, we could also have some time we meet face-to-face."*

(Nurse, Mugina Health Centre, KII).

## Discussion

We collected nurses' and midwives' perspectives on their capacity building needs in EmONC and how technology-enhanced learning can be maximized to bolster CPD trainings for nurses and midwives working in remote areas about EmONC. We also learnt from nurses' and midwives' perspectives the facilitators of and constraints that are mostly likely to influence the application of technology enhanced learning approach in training nurses and midwives working in remote areas about EmONC.

We found that access of CPDs by nurses and midwives still need to be improved because they still do not get all the needed EmONC skills based on their personal work experiences. Our study found that there is a need to provide refresher trainings about maternal and neonatal complications that may occur before, during, and after childbirth. The findings from this study corroborate those of studies conducted from different low resources countries across the globe [18–21], that established the need for providing adequate and regular trainings on EmONC for nurses at primary health level to enable them recognize the risk factors of pre-eclampsia, and be able to confirm it so that prophylactic interventions to manage it are initiated earlier. This finding also supports the results of a similar study conducted in Rwanda that highlighted lack of regular and continuous trainings for nurses and midwives about EmONC as a constraint to managing emergency maternal and neonatal complications [21].

The current study established that most of the EmONC trainings are still provided face-to-face and the use of technology enhanced learning approaches have not yet been embraced in delivering EmONC CPDs for nurses and midwives in remote areas. There are several explanations for this result. First, this may be due to individual and systemic factors including the limited health providers' familiarity with the internet and computer usage as well as low familiarity with online methods of teaching learning. Second, systemic factors such as limited IT resources at some remote health facilities [15, 22], and their readiness to adopt online learning approaches in building the capacity of their staff. This has also been confirmed by a study that involved managers of health facilities that identified challenges such as the lack of access to digital devices, poor or lack of internet access, poor online learning design, low digital skills of healthcare professionals, lack of time dedicated to online learning, and heavy workload of staff as barriers affecting the provision of CPD and capacity building trainings for health professionals using technology enhanced learning approach [15].

Our study found that nurses and midwives are willing to undertake EmONC CPDs delivered through blended approach using both online approach and hands-on-skills physical trainings. Our findings support those of a study conducted by Nyiringango et al. in which nurses, midwives, and physicians perceived the need for blended mode of CPD provision [23]. Furthermore, The results of our study provide additional support to the conclusions drawn from a comprehensive analysis that encompassed 24 randomized controlled trials conducted

in developed nations [24]. This review revealed compelling evidence that a combination of e-learning and traditional training methods, known as blended e-learning, yields superior outcomes in terms of enhancing attitudes and behaviors related to evidence-based healthcare compared to either face-to-face training or pure e-learning alone [24]. The blended learning approach can be extended to delivering CPDs programs on EmONC in remote areas, thereby addressing the challenges posed by geographical constraints and also fostering the quality of nurses and midwives through simulation and demonstration sessions [25].

Nurses and midwives found the first developed prototype of smartphone app training of the EmONC acceptable as it met the midwives' expectations in terms of the knowledge and skills' gap in EmONC. This finding is consistent with perspectives expressed by midwives in Tanzania and Democratic Republic of Congo about the effects of smartphone application in equipping them with knowledge useful for their practice of EmONC provision [26, 27]. In spite of the positive perspectives about the acceptability of the newly developed smartphone app, our study identified a number of constraints that may affect nurses' and midwives' utilization of the newly developed smartphone application. Nurses and midwives mentioned that the quality of and availability of internet in the remote areas, affordability of internet bundles, and the coordination of the online training for CPD purposes may affect the use of this application for CPD purposes. Like in other places where smartphone applications were piloted to provide trainings such as Tanzania, similar constraints above mentioned have been documented particularly lack of free internet access in maternity [26]. In addition to these challenges, unlike our study, the study from Tanzania further highlighted lack of time to read due to busy workloads [26]. To mitigate the issue of limited internet network access, nurses and midwives suggested that they actively use the app to access CPDs on EmONC if an offline version of the app is availed. Our study further suggested that regular monitoring to check the utilization of the app will need to be carried out to support nurses and midwives who may experience difficulties whilst using the app.

## Limitations

The current study has some limitations that may call for cautious interpretation of the findings. Participants were asked their views about the feasibility and acceptability of the application they have seen through a brief demonstration but which they have not used before. For this reason, we could not garner participants' views on the ease and/difficulty of using the application in accessing EmONC CPD materials. Could they have got enough time to test and use the newly developed application and accessing the content of the application, the study could have garnered more insights on the feasibility of the developed application. Another limitation is that data was collected virtually due Covid-19 pandemic containment measures. Although we tried our best to minimise any biases, the involvement of district based EmONC mentors to facilitate the smooth running of interviews may have led to some respondent biases.

## Conclusion

Strengthening the use of technology in delivering CPD on EmONC to nurses and midwives in remote areas of Rwanda is crucial. Currently, the utilization of online resources, such as academic websites, for self-learning among nurses and midwives is limited. Therefore, it is imperative to implement interventions that focus on enhancing their digital health literacy, including skills in information searching, appraisal, and selection of relevant content. By doing so, their proficiency in EmONC can be improved, leading to better healthcare outcomes for mothers and newborns in remote healthcare facilities.

While technology-based training can significantly enhance the delivery of CPD in EmONC for nurses and midwives stationed in remote areas, policymakers should also consider the

implementation of blended training programs. These programs should combine remote training with face-to-face sessions to ensure that nurses acquire the necessary practical skills. It is important to take into account the workload and availability of nurses and midwives when designing remote training programs. Additionally, gathering feedback from nurses and midwives through practical sessions using the newly developed application is essential. This feedback should focus on evaluating the ease of use of the application, the relevance of the EmONC content available on the app and identifying any necessary improvements to address their specific needs in EmONC. Further research in this area is warranted to ensure the effectiveness and acceptance of technology-based training initiatives in remote healthcare settings.

## Supporting information

**S1 Checklist. Human participants research checklist.**
(DOCX)

**S1 File.**
(DOCX)

**S1 Appendix. Randomly selected facilities.**
(DOCX)

## Author Contributions

**Conceptualization:** Thierry Claudien Uhawenimana, Mathias Gakwerere.

**Data curation:** Thierry Claudien Uhawenimana, Mathias Gakwerere, Jean Pierre Ndayisenga.

**Investigation:** Jean Pierre Ndayisenga.

**Methodology:** Thierry Claudien Uhawenimana, Mathias Gakwerere, Anaclet Ngabonzima, Jean Pierre Ndayisenga.

**Validation:** Jean Pierre Ndayisenga.

**Writing – original draft:** Thierry Claudien Uhawenimana, Assumpta Yamuragiye, Florien Harindimana, Jean Pierre Ndayisenga.

**Writing – review & editing:** Thierry Claudien Uhawenimana, Mathias Gakwerere, Anaclet Ngabonzima, Assumpta Yamuragiye, Florien Harindimana, Jean Pierre Ndayisenga.

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
