## [Decision Letter · Decision Letter 0]

14 Nov 2023

PONE-D-23-26083Nurses and Midwives' Perspectives on Technology-Enhanced Learning and Continuous Professional Development on Emergency Obstetric and Neonatal Care in RwandaPLOS ONE

Dear Dr. UHAWENIMANA,

Thank you for submitting your manuscript to PLOS ONE. After careful consideration, we feel that it has merit but does not fully meet PLOS ONE’s publication criteria as it currently stands. Therefore, we invite you to submit a revised version of the manuscript that addresses the points raised during the review process.

Three reviewers have provided constructive feedback, indicating where substantive revisions are needed to meet journal criteria for publishing research of a high technical standard.

We look forward to receiving your revised manuscript.

Kind regards,

Hannah Tappis, DrPH, MPH

Academic Editor

PLOS ONE

 “Not applicable”

Reviewers' comments:

Reviewer's Responses to Questions

**Comments to the Author**

1. Is the manuscript technically sound, and do the data support the conclusions?

Reviewer #1: Yes

Reviewer #2: Partly

Reviewer #3: Partly

2. Has the statistical analysis been performed appropriately and rigorously? 

Reviewer #1: No

Reviewer #2: N/A

Reviewer #3: N/A

3. Have the authors made all data underlying the findings in their manuscript fully available?

Reviewer #1: Yes

Reviewer #2: Yes

Reviewer #3: Yes

4. Is the manuscript presented in an intelligible fashion and written in standard English?

Reviewer #1: Yes

Reviewer #2: Yes

Reviewer #3: Yes

5. Review Comments to the Author

Reviewer #1: Description of how objectives were achieved especially factors that may influence

the application of this training approach in the Rwandan context is not clear. Mentioned that the study is a mixed method and elsewhere in the manuscript was mentioned as qualitative, need some clarification, otherwise the study sounds scientific with novelty of the subject.

Reviewer #2: This is a great start on an article which can be of great value for those considering the introduction of e-learning approaches within the context of addressing maternal mortality in LMICs. Please see below some thoughts for consideration:

The introduction states: Furthermore, available research from Rwanda has focused on health facility managers’ perceptions on the use of e-learning in providing CPD for health providers’ capacity building purposes [20]. It would be helpful to revist this in the discussion and understand how health facility managers' perspectives overlap or differ from that of nurses, midwives and EmONC mentors.

Consider providing demographic information about the participants, average years of experience with EmONC, level of education, previous experience with technological applications. This can help contextualize the findings.

Please provide more detail on the coding process. How as the initial codebook developed? How many qualitative research experts coded each transcript? Did the qualitative research experts review and code together or independently? At final review, did the qualitative research experts discuss codes until achieving consensus?

Theme 1: Access to EmONC CPD. This appears be less about access, with the predominant theme being perceptions on efficacy of in-person versus online training. Would suggest reframing the intro, moving the part about access to the moderators section (theme 3) and focusing on nurses’ perceptions of in-person versus online training.

“The common finding among all interviewees is that they appreciated the organizers of these CPD trainings, as they included the practical sessions as reported by one of the respondents.” Similarly, this may be less about appreciating the organizers and more about appreciation and need for practical, in-person trainings for specific topics.

“Participants expressed ambivalent views concerning the approach that can be used to on-job EmONC CPD courses.” With the majority preferring face-to-face trainings, it appears their views are similar and not ambivalent.

Theme 2: Experience in using technology to access EmONC information. This theme is particularly brief. There may be value in revisiting the findings and assessing what can be added to this section. It may also be worth considering moving this to the moderators section, as it touches upon self-efficacy, digital literacy and health literacy.

Theme 3: Moderators of technology enhanced learning for EmONC CPD delivery. This section appears to be a mix of perceived benefits (access to video, audio, learning from work/no transport fees), needs (increase in the number of topics covered,) and moderators (staff motivation to enhance their knowledge and skills, accessibility: timing of live courses aligned with staff's free time/annual leave). Would suggest moving the benefits and needs to Theme 1: Perceptions of Online versus In-Person EmONC Trainings. Would suggest moving the challenges from Theme 1: insufficient skills in IT, lack of continued access to strong internet network to the moderators theme.

Discussion: financial benefits that may be associated with attending face to face trainings. Second, systemic factors: limited IT resources at the facilities and their readiness to adopt online learning approaches in building the capacity of their staff. It is unclear if these are findings of another study or this study. If this study, they appear to be new findings not previously stated.

Given the number of focus groups and interviews conducted, it would be advisable to review the findings and offer a more robust analysis. While EmONC mentors where included, the majority of results focus on nurses and midwives. What were their views and how did they compare with nurses and midwives? Were there any difference in perceptions of staff from

District Hospitals versus Health Centers and Health Posts?

Reviewer #3: Overall this is an interesting study with potentially useful evidence generated to enrich the body of knowledge.

However, the manuscript does not allow the reader to benefit from the study. To improve the manuscript the following suggestions may be useful:

1. The title of the manuscript does not reflect the contents of the paper or rather the contents are not focused enough to be reflected in the title. Therefore either revise the title or (and that would be my suggestion) revise the manuscript to sharpen it to clearly address the key words from the title.

2. Provide a clearer rationale for the research (rather than one which is alluded to indirectly) and Rwandan context (in as far as it related to the focus of the study presented in the manuscript).

3. Provide a clear explanation for the e-learning platform being subject to the assessment in the research. There is a mention of the app being a new tool used for a short time under Limitations but really more information needs to be included in the earlier part of the manuscript.

4. Provide more information on the study sites and respondents to allow the reader understand the methodology better.

5. Assuming the data is reach in information, it may be useful to critically assess the themes to draw out a more nuanced and specific focus for the findings. Some of the aspects appear in multiple themes and overall the Findings section lacks a clear message - it 'jumps' from issues with IT skills and internet access, preference for face-to-face trainings, specific topics needed for training, then covers the strengths and positives of using e-learning, and other points all of which can build a more comprehensive picture if re-organised into a clearer narrative by the authors.

6. The discussion lacks inputs from a wider selection of other studies, which would be helpful for rooting the research conducted in a wider body of knowledge.

7. Relating somewhat to the rationale, and the presentation of findings the Conclusions lack a more pragmatic focus, which would make the manuscript more useful for policy-makers and practitioners alike. Therefore it would be useful to sharpen the section to improve the reach the research may offer.

There are other specific comments listed below:

P2: ‘Two focus group discussions with EmONC mentor, (…)’? -- should this say Two focus group discussions with EmONC mentors, (…)’?

P3: ‘The major causes of these deaths are pregnant-related complications known as obstetric complications such severe bleeding and infections that occur after childbirth and abortion, (…)’ – should this say ‘The major causes of these deaths are pregnant-related complications known as obstetric complications such as severe bleeding and infections that occur after childbirth and abortion, (…)?

P3: ‘One of the targets for the third sustainable development goals (SDGs) to reduce worldwide maternal mortality ratio (MMR) to (…)’ – should this say ‘One of the targets for the third sustainable development goals (SDGs) is to reduce worldwide maternal mortality ratio (MMR) to (…)’?

P4: ‘Technology enhanced learning has a number of benefits. First, it can mitigate some service delivery gaps that occur when nurses and midwives attend face to face trainings [13, 14]. Second, those who receive face to face trainings may not find it easy to transfer the skills they earn from such trainings [13, 14]. Third, some of the people who benefit from the face-to-face trainings may leave their places of work leaving personnel gaps [13, 14]. Fourth, technology enhanced learning are efficient because they can reach many people at a reasonable cost and at a regular time and meeting the pace and convenience of nurses and midwives [13, 14]. Lastly, need to shift from traditional way of learning new skills to the new way of acquiring skills using technological tools at nurses’ and midwives’ reach; promoting efficient and effective use of resources and also bolstering their capacity in digital health [15].':

- Unclear: how does the observation that healthcare providers accessing face-to-face training can have problems with transferring skills relate to benefits of using technology enhances learning? Please elaborate.

- How do points 1 and 3 differ? It seems in both the point is that due to in person attendance there are workforce gaps affecting service delivery. Please amend to make this clearer or if the two are in fact the same, please rewrite.

- Also, these are limitations of in-person training, not benefits of technology-enhanced learning - please make the point more clearly.

- The final sentence of the paragraph is not grammatically correct and therefore not very clear - please revise.

P4: ‘In particular, e-learning and remote training is cost-effective compared to traditional learning methods and can provide space for easy scale-up and improvement of content availability [16].’: Is this not a benefit of e-learning technologies discussed in the above paragraph? Please consider including in the list above.

P4/5: ‘Moreover, e-learning has proven its relevance in teaching and learning process amidst COVID-19 pandemic, including during challenging times of movement restrictions and lockdowns[17].’ – as above.

P5: ‘Despite the progress made, less is known about how remote nurses and midwives are facilitated to attend CPDs. It is in this regard that the current study sought to test the feasibility of providing CPDs to nurses and midwives in hard-to-reach health facilities in Rwanda using user-friendly phone-based technology.’ And onwards: The Rwandan context requires some revising - while relevant points building the framework for where this study fits are provided, the specifics of the Rwandan MNH system and HCP training within that require more information to help the reader (who we should assume will have limited knowledge of it) understand the topic.

P6: ‘Therefore, this study sought to understand from the nurses’ and midwives’ perspectives how technology-enhanced learning can be used to provide learner-centred CPD trainings for nurses and midwives about EmONC and factors that may influence the application of this training approach from the Rwandan context based on the newly developed EmONC smartphone application prototype.’: Overall the background assumes prior knowledge of the Rwandan context and 'jumps' between subtopics relevant as context to justify the need for the study. While potentially its benefits are understandable this section should be revised to build a clear case for the research while allowing the reader to learn more about the specific aspects of the healthcare system in Rwanda which make it interesting and relevant to focus on.

P6: Methodology:

- Dates when the study was conducted are missing - please include.

- This training programme / e-learning platform needs an introduction in the background section.

P6: ‘Data was collected from nurses and midwives from forty (40) public health facilities nationwide.’: How were these selected? How representative is the sample (if at all)? Out of how many overall? What type of facilities were included: lower/higher level, urban/rural, public/private, geographical spread, etc. Were the selected facilities part of a particular programme? Please include this information. This is somewhat addressed under Sample size and sampling strategy below, but it may be useful to reorganise the information to make it clearer.

P7: ‘In total, 54 individuals were included in this study.’: What was the response rate? Were the same respondents in the FGDs and KIIs or was any individual only asked to participate in group discussions or interviews?

P7: Data collection process: What was the difference between the FGDs and KIIs data collection tools? Did they cover the same issues?

P8: ‘We ensured that HCPs) felt (…)’ – please delete the superfluous bracket sign.

P8: ‘A thematic analysis was conducted and all five steps of qualitative-data analysis (reading, interpreting, coding, reducing, and displaying) were performed to ensure consistency within the data.’ - This sounds like steps from 'framework analysis' (Richie and Spencer) - is that the case?

P10 and P13: ‘ptototype’ – please correct to ‘prototype’.

P10: Findings: In general, please revise the findings to make the arguments you are trying to convey clear - these also need to be substantiated by the data by including a quote or providing a feel for how common (or uncommon) a particular point was, and/or whether it was voiced unanimously or a particular group of respondents found it more relevant than others.

P10: Theme 1: Access to EmONC CPDs: There are two points re e-learning courses but they are not presented as such:

1. Online courses are more challenging to access due to required IT skills and poor Internet infrastructure;

2. Points awarded for e-learning being insufficient in comparison to the effort required for completing the training.

P10, P12 and elsewhere: please standardise the spelling of ‘face to face’ as either ‘face to face’ or ‘face-to-face’ as both forms are used in the manuscript.

P 10: HMS and HBB:– Please spell out the abbreviations and check for any other abbreviations which may not be explained.

P11: ‘on-job EmONC’ should read ‘on-the-job EmONC’.

P11: ‘The respondents mentioned that despite the usefulness of online CPD trainings particularly during emergencies such as Covid-19 pandemic, in-person trainings were perceived effective in equipping nurses and midwives with knowledge and translational practical skills in EmONC as illustrated in this quote.’: In more general terms the role online trainings in a CPD programme and how they could be improved to provide a more useful resource for healthcare providers is also covered in Theme 3 suggestion there is a link between sub-sections that may be better explored in the manuscript.

P11: ‘However, some nurses and midwives; particularly (…)’ – should this read ‘However, some nurses and midwives, particularly (…)’?

P12: Theme 3: Moderators of technology enhanced learning for EmONC CPD delivery: The point on IT skills was raised previously - could these be linked or combined?

P13: ‘(…) it is difficult for us". One mentor mentioned.’ and ‘This is saddening to lose a mother or a baby." One midwife mentioned during the KIIs.’ - Standardise quotes: the follow up on one mentor mentioned coming after the quote does not follow the style used elsewhere.

P14: ‘During the discussions, nurses and midwives proposed to also avail an offline option of this prototype in order to mitigate the connectivity issue that may affect its functionality. Another barrier participants reported was financial constraint to access internet bundles and smartphones for some nurses and midwives, which can reduce their interest in using the application to learn.’: Internet access (as well as IT skills) have been raised earlier. In general, while the 'themes' cover specific topics, there is some overlap - whether that is a matter for reorganising information under each theme or providing clearer links this needs to be addressed to better guide the reader.

P15: ‘Respondent said: (…)’ – should this read ‘One respondent said: (…)’? Please correct formatting to non-cursive for not-quote.

P15: Discussion: The discussion feels somewhat lacking in evidence from a wider body of literature. Please consider reading more broadly about online learning programmes, platforms and tools which may provide a helpful caparison for your research findings.

P15: ‘Particularly, our study found that there is a need to provide refresher trainings about the management of pre-eclampsia.’: In fairness the focus on this particular topic does not come as strongly from the findings; eclampsia is mentioned alongside other topics flagged for future training ideas. If this was a particular critical point, please stress that in the findings.

P16: ‘Nurses and midwives are willing to undertaken (…)’ – should this read ‘Nurses and midwives are willing to undertake (…)’?

P16: ‘Nurses and midwives are willing to undertaken EmONC CPDs delivered through blended approach using both online approach and hands-on-skills physical trainings.’: This is a critical statement and I wonder whether this can be used better for fine-tuning the Findings section. The use of online training intuitively seems to be in-line with advances in technology and costs of in-person training but there are skills which cannot be learned by a phone app. Therefore it would be useful to understand from your study how the HCPs felt about e-learning in general, in terms of the specific app that they were using at present and how they would wish to see the use of e-learning going forward. What did they see as advantages, disadvantages and what the suggestions for making the platform better (more useful) were. This would provided a stronger analytical angle for policy and practice to draw out more specifically what works, what needs to improve and what the next steps ought to be.

P17: ‘Could they have got enough time to test and use the newly developed application and accessing the content of the application, the study could have garnered more insights on the feasibility of the developed application.’: This is a major point and really ought to have been raised as part of the introduction to the app to clarify exactly what was being used as a reference for the e-learning study!

P17: ‘Lastly, although the current study sought to explore how online and computer-based CPD delivery can be promoted to provide CPDs to nurses and midwives, it cannot be ignored that nurses and midwives need hand on skills practice to provide EmONC to the mothers and newborns hence a blended approach would always add value as highlighted by some respondents in this study.’: I am not sure this is a study limitation. Instead I think the context for the research needs to be set out more clearly to a priory specify that the nature of the practice of maternity staff requires practical skills which require in-person learning and practice and inline learning cannot be seen as a substitute for that. Instead the role of online training is to support, complement and enhance the learning process and as part of the study your findings have flagged what aspects make it a useful platform and where the shortcomings are. This could then be used by policy-makers to consider how to address the need for ongoing training (CPD) for maternity staff based on your work to incorporate what the evidence suggests should be continued, what should be developed further and what needs further thinking. This will make your research more useful for planning and informing policy and practice.

P18 onwards: References: Spelling of WHO to be standardised in the Bibliography.

6. PLOS authors have the option to publish the peer review history of their article (what does this mean?). If published, this will include your full peer review and any attached files.

Reviewer #1: **Yes: **Theoneste Hakizimana

Reviewer #2: No

Reviewer #3: No

---

## [Author Response · Author response to Decision Letter 0]

9 Jan 2024

On behalf of my co-authors, we appreciate your commentary and we hope that your continual review will enable us have our work improved and published.

---

## [Decision Letter · Decision Letter 1]

16 Feb 2024

PONE-D-23-26083R1Utilization of technology to provide on-job trainings on Emergency Obstetric and Neonatal Care: Perspectives of nurses and midwives working in Rwanda’s remote health facilitiesPLOS ONE

Dear Dr. UHAWENIMANA,

Thank you for submitting your manuscript to PLOS ONE. After careful consideration, we feel that it has merit but does not fully meet PLOS ONE’s publication criteria as it currently stands. Therefore, we invite you to submit a revised version of the manuscript that addresses the points raised during the review process.

Kind regards,

Hannah Tappis, DrPH, MPH

Academic Editor

PLOS ONE

Journal Requirements:

Reviewers' comments:

Reviewer's Responses to Questions

**Comments to the Author**

1. If the authors have adequately addressed your comments raised in a previous round of review and you feel that this manuscript is now acceptable for publication, you may indicate that here to bypass the “Comments to the Author” section, enter your conflict of interest statement in the “Confidential to Editor” section, and submit your "Accept" recommendation.

Reviewer #2: (No Response)

Reviewer #3: All comments have been addressed

2. Is the manuscript technically sound, and do the data support the conclusions?

Reviewer #2: Yes

Reviewer #3: Yes

3. Has the statistical analysis been performed appropriately and rigorously? 

Reviewer #2: N/A

Reviewer #3: N/A

4. Have the authors made all data underlying the findings in their manuscript fully available?

Reviewer #2: Yes

Reviewer #3: Yes

5. Is the manuscript presented in an intelligible fashion and written in standard English?

Reviewer #2: Yes

Reviewer #3: Yes

6. Review Comments to the Author

Reviewer #2: Write out full form of CPD at first use (abstract background)

Theme 1: Access to EmONC CPD. This appears be less about access, with the predominant theme being perceptions on efficacy of in-person versus online training. Access to CPD credits and internet is mentioned in the introduction, but not supported by quotes. If there are quotes supporting challenges in access, create a separate theme or consider adding to theme 3: Moderators of technology enhanced learning.

Theme 3: The middle section (participants expressed that the number of topics they covered regarding EmONC was not sufficient) does not appear to be a moderator and is more related to acceptability (theme 4).

Reviewer #3: Thank you for the revised version of the manuscript. It has benefited from the updates greatly and reads very well.

I have no major suggestions but some minor revisions are still necessary, but these are mainly relating to editing and formatting of the documents:

- As previously raised, on-job should be replaced by on-the-job;

- As previously raised, please decide on the writing of ‘face to face’ and keep the consistency as either face-to-face or face to face;

- Quotations need to be standardised – this has already been done but in one instance the quote does not follow the format (quote ‘Personally, as a midwife, (…)’), but also the way the quotes are presented could benefit from using a common style for indicating who the respondent was – normally after the quote you may see details such as profession, perhaps another identifier such as location (though ensuring this does not compromise the anonymity so perhaps a wider geographical region) and source of information (FGD, IDI) – with the relevant information in brackets immediately after the quote, instead of using text to describe the person as is currently applied. The quote would then read for example “xxxxxx” (Nurse, Y District, FGD);

- ‘Whatever complications that affect an expectant women experiences also causes serious effects to the baby’ – this reads somewhat clumsy – please revise;

- ‘This was done in order to find out if the newly design remote training tool (…)’ – should read ‘This was done in order to find out if the newly designed remote training tool (…)’;

- ‘Qualitative Research Experts spelt also as Qualitative Research experts – please standardise and consider using lower case lettering for the term;

- Use of numbering (1 and 2) or letters (i and ii) to be applied for listing of themes – at present the first one is listed under ‘1’ and the second under ‘ii’. Also, the introduction to themes lists two but in the findings there are three themes, which do not match the ones provided under ‘Data analysis’ - please revise to make sure the text is correct;

- As previously raised, flagging of pre-eclampsia under discussion does not match the findings; the explanation provided in the rebuttal letter needs to make its was into the manuscript to provide an explanation for why this particular aspect of emergency obstetric care is highlighted;

- There is no need to spell out EmONC after its first appearance in the text and use of the abbreviation alone is sufficient;

- ‘Supplementary information 2’: The information in the supplementary file may need revision to ensure the files are clearly labelled, but also in the current document whatever is included under header ‘1.1.1’ and ‘Table 1’ needs relabelling as these numbers do not seem to match the document. Additionally, while providing data on the general population of the respondents is useful given the lack of similar data on the sample in the research study presented in the manuscript, the information needs to be suitably adapted to the manuscript; instead it reads as a copy from a general report which makes it challenging to follow as a reader – please revise to make it clearer.

7. PLOS authors have the option to publish the peer review history of their article (what does this mean?). If published, this will include your full peer review and any attached files.

Reviewer #2: No

Reviewer #3: No

---

## [Author Response · Author response to Decision Letter 1]

20 Feb 2024

Regarding the issue raised in supplementary information 2, the authors have have acknowledged in the manuscript the limitation of not including the sample characteristics for the qualitative component. However, sample characteristics of the broader project that contributed to the data provided in this manuscript have been presented. Other minor issues are addressed in the manuscript and for details, see the rebutal letter.

---

## [Editor Report · Decision Letter 2]

26 Feb 2024

Utilization of technology to provide on-the-job trainings on Emergency Obstetric and Neonatal Care: Perspectives of nurses and midwives working in Rwanda’s remote health facilities

PONE-D-23-26083R2

Dear Dr. UHAWENIMANA,

We’re pleased to inform you that your manuscript has been judged scientifically suitable for publication and will be formally accepted for publication once it meets all outstanding technical requirements.

Kind regards,

Hannah Tappis, DrPH, MPH

Academic Editor

PLOS ONE